# Audit and Feedback in the Hospitals of the Emergency Networks in the Lazio Region, Italy: A Cross-Sectional Evaluation of the State of Implementation

**DOI:** 10.3390/healthcare11010024

**Published:** 2022-12-22

**Authors:** Carmen Angioletti, Luigi Pinnarelli, Paola Colais, Laura Angelici, Egidio de Mattia, Marina Davoli, Antonio Giulio de Belvis, Nera Agabiti, Anna Acampora

**Affiliations:** 1Department of Epidemiology, Regional Health Service Lazio Region, 00147 Rome, Italy; 2Critical Pathways and Evaluation Outcome Unit, Fondazione Policlinico Universitario “A. Gemelli”—IRCCS, 00168 Rome, Italy; 3Scuola Superiore Sant’Anna, 56127 Pisa, Italy; 4Università Cattolica del Sacro Cuore, 00168 Rome, Italy

**Keywords:** audit and feedback, quality improvement, emergency care

## Abstract

Audit and Feedback (A&F) is an effective multidimensional strategy for improving the quality of care. The optimal methods for its implementation remain unclear. This study aimed to map the state of art of A&F strategies in the hospitals involved in a time-dependent emergency network. For these purposes, a structured questionnaire was defined and discussed within the research group. This consists of 29 questions in three sections: (1) characteristics of the structure, (2) internal feedback systems, and (3) external feedback systems. All structures involved in the network were invited to participate in the e-survey by indicating a Health Management representative and a clinical representative for the Cardiovascular (CaV) and/or for the Cerebrovascular area (CeV). Of 20 structures invited, a total of 13 (65%) responded to the survey, 11 for the CaV area and 8 for the CeV area. A total of 10 of 11 (91%) facilities for the CaV area and 8/11 (75%) for the CeV area reported that they perform A&F activities. All facilities perform at least one of the activities defined as “assimilating A&F procedures.” The most frequent is the presentation and discussion of clinical cases (82% CaV and 88% CeV) and the least is the identification of responsible for improvement actions (45% CaV and 38% CeV). In 4/10 (40%) facilities for the CaV area and 4/8 (50%) for the CEV area, corrective actions are suggested or planned when the feedback is returned. These results confirm the need to define, in a synergistic way with the relevant stakeholders, an effective and agreed A&F intervention to improve the level of implementation of A&F strategies.

## 1. Introduction

Conducting regular assessment and providing feedback is one of the “Seven ways to improve quality and safety in the hospital” proposed by the DUQuE Project [1]. Audit and feedback (A&F) is defined as “any summary (written or verbal) of clinical performance of health care over a specified period of time” [2]. The purpose of A&F is to measure clinician’s performance, to compare it to a standard, and then to feed the results back to the clinician to improve practice. A&F are commonly used to help providers to identify the gap between knowledge and practice and improve quality of care; the assumption is that professionals will improve their performance when feedback demonstrates deficiencies in process or outcomes of care. According to two Cochrane reviews [2,3], A&F activities generally lead to small but potentially important improvements in professional practice. Therefore, audit and feedback are key quality improvement strategies, which can be applied individually or as part of multifaceted interventions. Audit and systematic monitoring need to be embedded in departmental quality management mechanisms, with all professionals participating and receiving feedback on performance. Despite the high prevalence of A&F as a quality improvement strategy, the optimum methods for implementation of such interventions and the characteristics of A&F that lead to greater impact are unknown [4,5,6,7]. EASY-NET—“Effectiveness of Audit & Feedback strategies to improve healthcare practice and equity in various clinical and organizational settings”—is a multiregional Italian research program aiming at comparing different ways to conduct A&F to improve healthcare quality for several health conditions and in different geographical and organizational contexts in seven Italian regions including Lazio. Within this program, the Lazio region (program’s coordinator and responsible for the Work Package 1), is experimenting the implementation of A&F to improve emergency care for two acute conditions, acute myocardial infarction (AMI) and ischemic stroke involving voluntarily adherent hospitals. To face the complexity of acute conditions management, such as AMI and ischemic stroke, the Lazio Region implemented an emergency network involving different healthcare providers—i.e., emergency medical services (EMSs) and hospitals. The entire regional area has been divided into different sub-areas. For each sub-area a reference hospital (HUB) has been identified to which other hospitals (SPOKEs) must refer. Furthermore, starting from 2005, a regional program aimed at evaluating healthcare outcomes—called P.Re.Val.E. (Programma Regionale Valutazione Esiti)—is publicly available. The P.Re.Val.E. regularly calculate and publish the results of 159 outcome/process indicators encompassing many different clinical areas at both hospital and community level using a dedicated web platform. Throughout the platform, it is also possible to request the activation of an audit and feedback procedure. The evaluated healthcare pathways also include emergency care for AMI and ischemic stroke [8]. Within this framework, as part of the research activities, the EASY-NET Lazio WP1 (WG) intended to integrate experimental A&F strategies with existing initiatives at hospitals and regional level (such as the mentioned P.Re.Val.E.) with the final goal to improve healthcare quality. In this regard, the aim of the present research was to map the ongoing state of implementation of A&F strategies in the hospitals working into the emergency network of the Lazio region.

## 2. Materials and Methods

To map the state of implementation of A&F strategies in the hospitals involved in the EASY-NET Lazio project, a baseline survey was set up and administered.

### 2.1. Survey Development

At first, to outline the A&F definition and implementation strategies in time-dependent conditions management, a review of the scientific and grey literature was performed. Studies describing or experimenting with Audit and Feedback interventions conducted in emergency/urgency settings either exclusively or as a multidisciplinary intervention were collected using scientific and generic search engines (Pubmed, Google Scholar). The optimal elements of an audit and feedback intervention were extracted from the retrieved studies and grouped according to the underlying concept. Starting from the items identified from the reference literature, a structured questionnaire was defined. In order to evaluate its completeness, clarity, and intelligibility, a preliminary version of the questionnaire was discussed among three Authors of the research group (AC, PL, CP) and later revised by two additional Authors (AA, AL). Where appropriate, a wording revision was made. 

### 2.2. Survey Administration

All the hospitals involved in the emergency network of the Lazio region were invited to participate in the survey by indicating a health management representative and a clinical representative for the cardiovascular area and/or for the cerebrovascular area. If a hospital participates in both the investigated areas, answers were collected separately. The questionnaire was electronically distributed through the Redcap instrument (REDCap 2014) and the answers were automatically collected. A series of reminders were sent in order to increase the participation that was on a voluntary base. Cardiovascular and cerebrovascular area were analyzed separately. Descriptive statistics were performed using frequencies and percentages. Results were graphically reported through pie and bar charts. Data were analyzed using Microsoft^®^ Excel (2016).

## 3. Results

### 3.1. Survey Development

Three reference studies were adopted to draft the questionnaire [2,4,9,10,11]. Based on the results of the mentioned evidence, the following dimensions were investigated: (a) different levels of responsibility and involvement; (b) different recipients (e.g., individual or group); (c) different sources of feedback (e.g., supervisor, senior colleague, professional standards review organization, etc.); (d) different frequencies, durations, and content; (e) linkage to economic incentives or reimbursement schemes [2]; (e) educational materials; (f) multifaceted interventions (combination of two or more interventions) [4]; (f) action plans/coping strategies; *(*g) goal setting; and (h) nature of the data [9]. The final version of the questionnaire includes twenty-eight items organized into three sections according to the underlying concept: (1)Audit and feedback activities (2)Internal feedback systems (3)External feedback systems

The investigated items in the different sections are showed in Table 1 (Section 3.1).

### 3.2. Survey Administration

Overall, 20 hospitals were invited. Of these, 13 (65%) responded to the survey: 11 units responded for the CaV area, 8 for the CeV area, and 6 for both areas. The total number of participating units was 19. For the CaV area, answers were collected from nine Hubs and two Spokes. One of the hub hospitals is a semi-Governmental Teaching Research Hospitals and the other eight are general public hospitals. The participating two spokes are both general public hospitals. For the CeV area, answers were collected from three Hubs and five Spokes. Three Hubs are Teaching Research Hospitals, two Public and one semi-Governmental, and the five Spokes are all general public hospitals.

#### 3.2.1. Audit and Feedback in the Cardiovascular Area

Out of the 11 hospitals that participated in the survey for the CaV area, 10 (91%) reported performing A&F activities at unit level and 8 (73%) declared the presence of a hospital unit dedicated to their implementation (e.g., Quality Management Unit) (Figure 1). All facilities declared to perform at least one of the activities defined as “assimilating A&F procedures.” Among these, the most frequently reported activity was the presentation and discussion of clinical cases (82%) and the least frequent was the identification of those responsible for the implementation of improvement actions (45%).

Regarding the internal feedback systems, 10 respondents (91%) declared to have an internal reporting system, but 6 of these (60%) stated that the reporting system does not clearly indicate the specific behavior to change. Only four of these (40%) declared that corrective actions are suggested or planned when the feedback is returned. 

The periodicity of the reporting system release resulted heterogeneous: five respondents (50%) said they receive it quarterly, while two (20%) said they receive it monthly, and two others (20%) on an annual basis. 

Regarding the information in the reporting system, seven respondents (70%) reported that data are released in an aggregate manner, while three (30%) reported receiving data at both aggregate and individual patient level. Six respondents (60%) usually receive reports including comparison with benchmarks/standards. In addition, regarding external feedback systems, nine out of the eleven (82%) respondents for the CaV area reported to regularly consult the indicators published by the regional program for outcomes evaluation (P.Re.Val.E) to monitor their performance. Regarding the quality control and completeness of data recorded in Health Information Systems, ten respondents (91%) reported being equipped with systematic procedures to achieve this goal. Specifically, seven out of ten respondents (70%) reported having a dedicated Unit to implement such systematic procedures. Finally, six respondents (55%) reported how P.Re.Val.E. indicators are used as feedback systems associated with a reward or incentive system (e.g., budget targets).

#### 3.2.2. Audit and Feedback in Cerebrovascular Area

All the eight hospitals (100%) that responded to the survey for the CeV area declared they had carried out A&F, and six of these (75%) declared that they have a dedicated Unit for the implementation of A&F activities (Figure 2). All of these declared that they had performed at least one of the activities assimilated to A&F. The presentation and discussion of clinical cases (88%) was the activity most frequently performed and the identification of those responsible for the implementation of improvement actions was the least frequent (38%).

Regarding the internal feedback systems, all eight participants for the CeV area (100%) declared to have an internal reporting system, but six of these (75%) stated that the reporting system does not clearly indicate the specific behavior to change. Four of these (50%) declared that corrective actions are suggested or planned when the feedback is returned. 

The periodicity of the reporting system release resulted heterogeneous: four respondents (50%) said they receive it quarterly, while three (30%) said they receive it monthly. 

Regarding the information in the reporting system, six respondents (75%) reported that data are released in an aggregate manner, while two (25%) reported receiving data at both aggregate and individual patient level.

Four respondents (50%) usually receive reports including comparison with benchmark/standards. In addition, regarding external feedback systems, six out of the eight (75%) respondents for the CeV area reported to regularly consult the indicators published by the regional program for outcomes evaluation (P.Re.Val.E) to monitor their performance. 

Regarding the quality control and completeness of data recorded in Health Information Systems, seven respondents (88%) reported being equipped with systematic procedures to achieve this goal. Specifically, all the participants (100%) reported having a dedicated Unit to implement such systematic procedures.

Finally, three respondents (38%) reported how P.Re.Val.E. indicators are used as feedback systems associated with a reward or incentive system (e.g., budget targets).

## 4. Discussion

The present study aimed to explore and describe the state of the art in the implementation of Audit and Feedback (A&F) strategies for improving emergency care for AMI and ischemic stroke patients in the hospitals of the Lazio region, in Italy. To this end, a specific questionnaire, evaluating the presence of such a strategy and its characteristics, was defined and administered to representative health professionals working in clinical unit of the participating hospitals involved in the emergency networks (for AMI and ischemic stroke) of the Lazio region. The results from the present assessment showed high heterogeneity in the level of implementation of different activities belonging to A&F interventions across the 19 respondent units of the 13 participating hospitals. At first, the survey investigated the presence of such activities. Almost all of the respondents stated that A&F activities (or at least activities defined as “assimilating to A&F procedures”) are regularly carried out at the unit level and in most of the participating hospitals a medical unit dedicated to A&F implementation is present. A&F assimilating activities were evaluated according to a growing scale of complexity. In general, respondents from the cardiovascular unit reported all of the explored activities more frequently than cerebrovascular ones. Monitoring indicators are widely reported in the cardiovascular area, but in fewer cases, a report summarizing the measured performance is fed back to involved stakeholders. Instead, these activities are reported by about half of the respondent units for the cerebrovascular area. Organizing audit meetings, involving all stakeholders (specialists, nurses, other health professionals) and discussing indicators, is declared by about half of respondents from CaV units and in about one-third of CeV units. Although the presentation and discussion of clinical cases was the most reported assimilating activity both for the CaV area and for the CeV area, it is important to point out that in some cases, it is the unique activity reported or, in any way, not linked to critical results fed back. Although most respondents declared to receive periodic internal feedback, it is to note that few of these include a comparison with benchmark/standard values and in many cases the specific behaviors they intend to change along with a corrective action plan are not indicated. Finally, most participating hospitals for both CeV and CaV areas reported that they regularly consult the indicators published by the regional program for outcomes assessment (P.Re.Val.E) to monitor their performance. A&F strategies are, in general, widely used to improve professional practice, alone or as a component of multi-pronged quality improvement interventions [3]. The present results confirmed that also all the interviewed hospitals have implemented strategies assimilating to clinical audit and/or feedback activities. However, the main limitation of this study comes from the use of an electronic survey that limited the response rate to 65%, despite the numerous reminders provided either via e-mail or by telephonic contacts. This could introduce a possible response bias. In fact, hospitals where A&F implementation measures are in place might be more likely to participate in the survey, with the risk of overestimating the frequency of activities and feedback systems, which, in practice, might be lower and also more heterogeneous. 

Nowadays, it is well recognized that A&F has the potential to improve the quality of care starting from the concept that providing health professionals with objective data on their clinical performance may overcome their tendency to overestimate it and stimulate them to make efforts for improving their actions in the process of care [12,13]. Scientific literature in the field recognized that A&F interventions are most effective if the feedback is provided directly by the supervisor or by a respected colleague [3], then this evidence suggests that implementing a dedicated hospital unit responsible for carrying out these activities could be an effective strategy to improve A&F effectiveness. An additional feature that has been demonstrated to improve the efficacy of these interventions is the clear indication of specific goals to be reached and the specification of a consistent action plan including behaviors under the control of the recipients and easy to modify [3,9]. Recent evidence highlights that, frequently, the feedback is delivered by research teams conducting experimental evaluation and in a few cases by supervisors with a strong commitment and a clear indication of goals to be reached. These features could reduce the impact of these strategies on quality improvement [3]. Colquhoun et al. [9] have shown that theory is rarely invoked in the design of healthcare A&F interventions. Instead, most current A&F interventions appear to be guided by intuitive, non-theoretical ideas about what might work. Considering that it has been demonstrated by previous research that how an A&F intervention is designed may impact its effectiveness, current efforts should be oriented to optimize their design and delivery in order to amplify their impact in terms of care quality improvement [3]. Recently, Brehaut, and Colleagues [5] identified 15 key suggestions for designing and delivering effective practice feedback interventions, in which the recommendations or plan corrective actions are also included. The authors pointed out that, to be more effective, feedback must be: explicit, specific, time-limited, recipient-defined, and challenging but achievable. The first group of suggestions refers to the “nature of the feedback’s desired action”. As shown in this study, in most cases, the feedback reported a comparison between the recipient’s results and a standard or benchmark value but, in a few cases, respondents reported a clear specification of behaviors to change or corrective actions to plan. Regarding the “nature of the data available for feedback”, most respondents declared that feedback provides patients with data aggregated at the unit level Specific data, indeed, are demonstrated to be more effective [9]. Providing feedback periodically is another evidence-based suggestion. Most of the respondents reported that the feedback is periodically provided but with different timing (mainly every three months). The last part of the survey explored the level of consultation of the indicators published regularly by the regional program P.Re.Val.E. Most respondents reported using them for monitoring the unit performance but rarely these indicators are included in the budgeting process reports they receive, then efforts should be made in order to improve the dissemination and knowledge about this useful web tool. Giving recipients positive reinforcement could increase the effectiveness of the interventions encouraging recipients to improve their performance.

## 5. Conclusions

The presented results highlight the high heterogeneity in the implementation of A&F strategies to improve in-hospital emergency care for patients affected by time-dependent conditions such as Stroke and AMI. These findings confirm the need for the definition of a shared guidance for implementing these A&F strategies, by also involving professionals from the participating hospitals. The EASY-NET project has the great ambition to define a theory-oriented and evidence-based A&F intervention, to implement it, and to evaluate characteristics influencing its efficacy.

Furthermore, although the main activities of the project are focused on the in-hospital phase of the pathways dedicated to Stroke and AMI emergency management (from arrival in the ER to discharge), it also pays attention to the other phases of the aforementioned pathways, including the pre-hospital phase of rescue and transport of the patient to the hospital, and the post-discharge phase (30 days) [14].

Experimenting with different A&F interventions within the EASY-NET program could lead to a better understanding of how to correctly implement A&F strategies, which is essential to trigger real improvement actions in real-life clinical practice [15].

## Figures and Tables

**Figure 1 healthcare-11-00024-f001:**
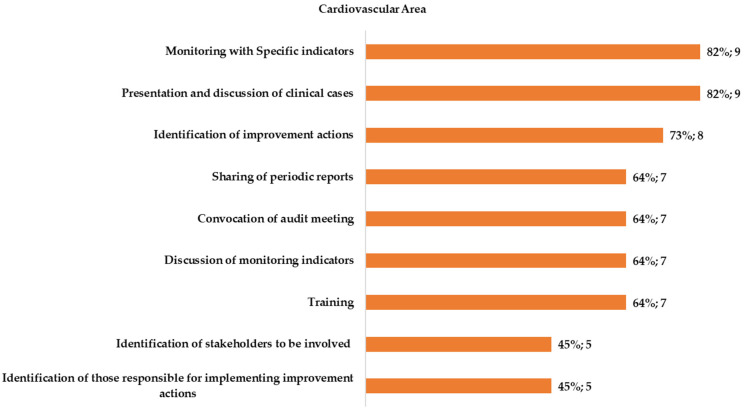
Type of assimilating A&F procedures reported by the respondents for the cardiovascular area.

**Figure 2 healthcare-11-00024-f002:**
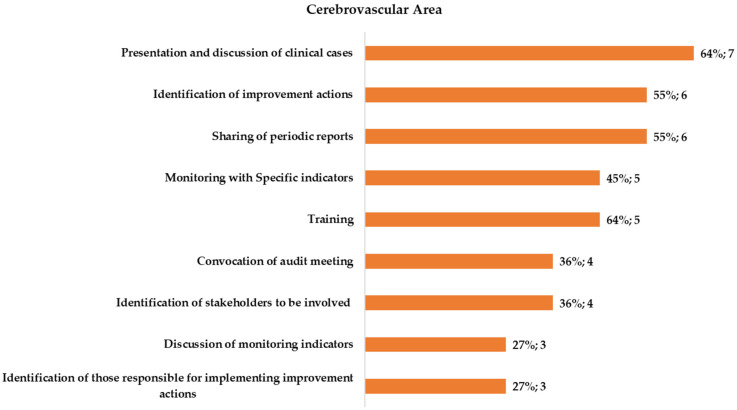
Type of assimilating A&F procedures reported by the respondents for the cerebrovascular area.

**Table 1 healthcare-11-00024-t001:** Investigated Audit and Feedback items.

Sections	Items
**Audit and feedback activities**	1. The presence of A&F procedures at unit level 1.1. The year of A&F activation 1.2. The presence of a specific unit/office dedicated to the A&F implementation 2. Implementation of “Assimilating A&F activities” at unit level - No activities - Periodic feedback report - Monitoring specific indicators - Stakeholder involvement (specialists, nurses, health management, other relevant professionals) - Audit meeting organization - Discussion of specific indicators - Discussion of clinical cases - Definition of improvement actions - Identification of one or more responsible for implementation of actions - Training 3. The presence of systematic procedures for the quality control of data flowing in HIS. 3.1. The year of data quality control activation 3.2. The presence of a specific unit/office dedicated to data quality control 4. Implementation of data quality control activities at unit level - Internal control of the discharge register (DR) Sample analysis of clinical chart and comparison with DR
**Internal feedback systems**	5. The presence of a periodic internal feedback system 5.1. The year of periodic feedback activation 5.2. The frequency of periodic feedback (e.g., monthly, every 3 months, six months, year, other) 5.3. The level of data aggregation (individual, aggregated, both) 5.4. The recipients for periodic feedback (single professionals, monodisciplinary, multidisciplinary or multi-professional teams, health management professionals) 5.5. Clear indication of the behaviors to change 5.6. Clear specification of an action plan for improvement 5.7. Clear indication of standards or benchmark for comparison 6. Indicators 6.1. List indicators 6.2. The year of availability of specific indicators 6.3. The presence of equity indicators 6.4. Equity dimensions considered 6.4.1. Availability of data for equity level calculation 6.4.2. List data sources 7. Additional suggested indicators
**External feedback systems**	8. The use of indicators from the regional program P.Re.Val.E. for monitoring 8.1. The year in which that use started 8.2. P.Re.Val.E. data quality control 8.3. Integration of P.Re.Val.E. indicators in the budgeting process

HIS—Health Information systems.

## Data Availability

Not applicable.

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
