# Peer review of "Audit and Feedback in the Hospitals of the Emergency Networks in the Lazio Region, Italy: A Cross-Sectional Evaluation of the State of Implementation"

_healthcare, 2022, doi:10.3390/healthcare11010024_

Round 1

Reviewer 1 Report

In the present paper, Angioletti et al described the state of the art of Audit and Feedback strategies implemented in the hospitals of the Lazio region involved in a time-dependent emergency network for AMI (Cardiovascular area, CaV) and stroke (Cerebrovascular area, CeV). 

For this purpose, authors developed a survey consisting of 29 questions, based on items identified in a review of scientific and grey literature, and electronically administered the survey to the 20 hospitals of the network. Of these, thirteen (65%) responded to the survey, prevalently from Hub hospitals for the CaV area. Results showed high heterogeneity in the level of implementation of different activities, even though all the interviewed hospitals have implemented strategies assimilating to clinical audit and/or feedback activities. 

This topic is undoubtedly relevant for the reader who is interested in implementation strategies to improve quality of care, especially in time-dependent settings.

The paper is well-written, methods for developing the questionnaire and for involving the hospitals of the network are accurately described, the descriptive analysis is detailed and well-conducted and results are well-reported.

Nevertheless, it presents some unavoidable limitations due to the on online survey methodology used, which are not mentioned neither discussed by authors. Even though the hospitals involved are those included in the P.Re.Val.E. regional program, and consequently are quite representative of regional hospitals where IMA and stroke are managed, only 65% responded to the survey, introducing a possible respondent bias. Hospitals where A&F implementation measures are in place may be more prone to respond to the survey, with the risk of overestimating the frequency of activities and feedback systems, which may in practice be lower and even more heterogeneous. Please make a comment on these limitations in the discussion section.

Author Response

The paper is well-written, methods for developing the questionnaire and for involving the hospitals of the network are accurately described, the descriptive analysis is detailed and well-conducted and results are well-reported.

Nevertheless, it presents some unavoidable limitations due to the on online survey methodology used, which are not mentioned neither discussed by authors. Even though the hospitals involved are those included in the P.Re.Val.E. regional program, and consequently are quite representative of regional hospitals where IMA and stroke are managed, only 65% responded to the survey, introducing a possible respondent bias. Hospitals where A&F implementation measures are in place may be more prone to respond to the survey, with the risk of overestimating the frequency of activities and feedback systems, which may in practice be lower and even more heterogeneous. Please make a comment on these limitations in the discussion section.

Response:

Dear Reviewer, thank you for your precious suggestion.

We included this limitation between lines 224 and 230 on page 8 as follow:

"However, the main limitation of this study comes from the use of an electronic survey that limited the response rate to 65%, despite the numerous reminders provided either via e-mail or by telephonic contacts. This could introduce a possible response bias. In fact, hospitals where A&F implementation measures are in place, might be more likely to participate in the survey, with the risk of overestimating the frequency of activities and feedback systems, which in practice might be lower and also more heterogeneous."

Reviewer 2 Report

1 The paper is very readable and fairly written. DOI numbers are missing, photos are unreadable.

2.The study is very interactive, showing current world problems.

3. original topic, brings innovation a different perspective.

4. The text is clear and easy to read.

5. coherent conclusions, presented very clearly 

Poorly legible caption above table 1

Please reword the conclusions

Please find more recent publications 

Arrange Figure 1 and Figure 2. from largest percentage to smallest percentage 

Author Response

Dear reviewer, thank you for your precious comments.

1 The paper is very readable and fairly written. DOI numbers are missing, photos are unreadable.

 Response 1: As rightly suggested, we have included the DOI numbers in the bibliography except for references 1, 2 which are guidebook and policy summary respectively.

2.The study is very interactive, showing current world problems. Thank you.

3. original topic, brings innovation a different perspective. Thank you.

4. The text is clear and easy to read. Thank you.

5. coherent conclusions, presented very clearly. Thank you.

Poorly legible caption above table 1. Thank you. The font has been enlarged (line 117, page 3).

Please reword the conclusions. Thank you for your suggestion. Conclusions have been reviewed (lines 272-286, pages 8-9)

Please find more recent publications. Thank you for your suggestion. Three new references (7, 10, and 11) have been incorporated.

Arrange Figure 1 and Figure 2. from largest percentage to smallest percentage. Thank you for your suggestion. Figure 1, 2 have been revised (pages 5,6)